# The Impact of Presentation Style on Human-In-The-Loop Detection of Algorithmic Bias

Po-Ming Law *
Georgia Institute of Technology

Sana Malik †
Adobe Research

Fan Du ‡
Adobe Research

Moumita Sinha §
Adobe Research

## ABSTRACT

While decision makers have begun to employ machine learning, machine learning models may make predictions that bias against certain demographic groups. Semi-automated bias detection tools often present reports of automatically-detected biases using a recommendation list or visual cues. However, there is a lack of guidance concerning which presentation style to use in what scenarios. We conducted a small lab study with 16 participants to investigate how presentation style might affect user behaviors in reviewing bias reports. Participants used both a prototype with a recommendation list and a prototype with visual cues for bias detection. We found that participants often wanted to investigate the performance measures that were not automatically detected as biases. Yet, when using the prototype with a recommendation list, they tended to give less consideration to such measures. Grounded in the findings, we propose information load and comprehensiveness as two axes for characterizing bias detection tasks and illustrate how the two axes could be adopted to reason about when to use a recommendation list or visual cues.

**Index Terms:** Computing methodologies—Machine learning; Human-centered computing—Empirical studies in HCI

## 1 INTRODUCTION

From financial services and recruitment to child welfare services and criminal justice, people have begun to employ machine learning (ML) systems to support decision making. In predictive policing, for example, police departments have utilized ML to predict crime hotspots and determine where to deploy officers [29]. Aside from assisting humans in making decisions, ML systems sometimes replace humans in the decision-making process. For instance, some financial institutions have employed ML to automate loan decisions [14].

Many ML researchers have warned that these systems may produce results that bias against certain demographic groups (e.g., ethnic and gender minorities) [20]. Such biases are often known as algorithmic bias [31]. Commonly-cited examples of algorithmic bias can be found in criminal justice and hiring. In some US states, judges use ML to assess defendants' risk of recommitting crimes when determining sentences [33]. Some journalists found that a risk assessment tool called COMPAS misclassified Black defendants as high risk twice as often as White defendants [6]. In hiring, recruiters often use ML to automatically sift through job applications [22]. These systems may reject job applicants from certain demographic groups much more frequently [35]. The discriminatory treatments can deprive under-represented groups of economic mobility and opportunity [7, 12].

ML modelers (i.e. practitioners who train ML models) can audit their models for algorithmic biases before model deployment. Many ML researchers have developed semi-automated bias detection tools (e.g., [5, 32, 34]). These tools often enumerate a set of performance measures across different demographic groups to identify and report the performance measures that indicate potential biases. For example, a bias detection tool may find that the classification accuracy on African Americans is significantly lower than the rest of the population and report the low accuracy. Users can then review the reported measures and apply their domain knowledge to determine if these measures imply real-life consequences.

A consideration in designing semi-automated bias detection tools concerns the presentation style of the reported measures. Two common presentation styles are a recommendation list (e.g., [5]) and visual cues (e.g., [32]). A recommendation list shows only the performance measures that indicate potential biases. The user interface can provide an option for examining other measures that were not reported to have biases. On the other hand, using visual cues, a user interface can show all the enumerated performance measures and highlight the ones that may indicate biases.

Yet, there is a lack of guidance concerning the decisions of which presentation style to use in semi-automated bias detection tools. Does the choice of a presentation style matter? Under what scenarios should a recommendation list or visual cues be used?

Investigating the considerations in choosing a presentation style entails understanding how users review reported performance measures. To explore user behaviors in the review of bias reports, we developed two semi-automated bias detection prototypes that differed in presentation style. The first presented reported measures in a recommendation list and the second used visual cues to highlight the reported measures. We conducted a small lab study in which 16 participants used both prototypes to review bias reports and select performance measures they wanted to investigate further.

Our analysis looked into the performance measures that are automatically reported as biases (hereafter, ***automatically-reported measures***), the measures that are not automatically reported by a bias detection tool as biases (hereafter, ***unreported measures***), and the measures that are selected by participants for further investigation (hereafter, ***manually-selected measures***). We found that participants often wanted to investigate unreported measures further. Yet, when examining a recommendation list, participants appeared to consider the unreported measures less even when they were given an option to examine such measures. The results implied that although some participants considered unreported measures important, they tended to be less comprehensive in reviewing these measures when the interface employed a recommendation list.

Grounded in the findings, we propose two axes—an information load axis and a comprehensiveness axis—to characterize bias detection tasks. The characterization enables us to reason about which presentation style to use in what scenarios. We argue that when comprehensiveness in bias detection is not a priority (low comprehensiveness) and when there are many performance measures for user review (high information load), recommendation lists could be used for presenting automatically-reported measures. In contrast, when comprehensiveness is a concern (high comprehensiveness) but when there are few performance measures for review (low information load), visual cues could be employed. Based on the two axes, we discuss high comprehensiveness and high information load as an

*e-mail: pmlaw@gatech.edu

†e-mail: sana.malik@adobe.com

‡e-mail: fdu@adobe.com

§e-mail: mousinha@adobe.com

Graphics Interface Conference 2020
28-29 May

under-examined research area.

The contributions of this paper are two-fold: **1)** findings from a lab study that investigated the impact of presentation style on how users review a bias report and **2)** the information load and comprehensiveness axes for considering what presentation style to apply when designing semi-automated bias detection tools. In the field of algorithmic bias, the investigations of the influence of interface design on human-in-the-loop bias detection have been scarce [26]. We contribute one of the first few studies to show that interface design holds power to affect the biases users find. We hope that our findings will raise awareness of the importance of making cautious design decisions when creating semi-automated bias detection tools and encourage researchers to explore other facets of design that might affect user behaviors in bias detection.

## 2 BACKGROUND AND RELATED WORK

Here, we summarize research in the definitions of algorithmic bias and semi-automated bias detection tools.

### 2.1 What is Algorithmic Bias?

The ML community has proposed various types of biases in ML models [15,19,23–25,39]. Our focus is a specific kind of algorithmic bias called group biases. Rovatsos et al. [31] offered a comprehensive review of such biases. They defined group biases as "the unfair treatment of a group (e.g., an ethnic minority, gender or type of worker) that can result from the use of an algorithm to support decision making." They commented that group biases are highly related to discrimination and fairness. Group biases may imply discrimination against individuals based on some protected characteristics. The UK Equality Act 2010 identified nine protected characteristics, including age, disability, sex, and religion [2]. Regarding fairness, Rovatsos et al. [31] identified two forms: procedural fairness and outcome fairness. Procedural fairness means that an algorithm "processes data about individuals in the same way, regardless of characteristics such as gender and ethnicity" while outcome fairness concerns how equally the outcomes of a decision-making process "are distributed across individuals and social groups within the population." Group biases imply that an ML algorithm is unfair either in its procedure or in the outcomes it produces.

These definitions of group biases are high-level and general. To provide more concrete definitions of group biases in ML models, some ML researchers defined biases statistically. They proposed bias measures to automatically detect the presence of group biases [19, 23, 39]—a model has biases if the bias measures say so. Computing a bias measure often involves measuring model performance on two demographic groups and checking if the two performance measures are different. When there are more than two demographic groups (e.g., in the case of ethnicity), a reference group is often defined [17, 32]. The performance measure of each group can then be compared with that of the reference group to compute the bias measure. A large difference indicates group biases.

Common bias measures include accuracy disparity [18], demographic parity [16], disparate mistreatment [39] and equalized odds [19]. Accuracy disparity compares the classification accuracies between demographic groups. Demographic parity compares classification rates between demographic groups while disparate mistreatment and equalized odds determine if there is a difference in true and false positive rates between groups.

In this paper, we adopt a statistical definition of group biases: A group bias exists if a bias measure indicates a statistically significant difference in the performance measures between demographic groups. We consider the four bias measures mentioned above: accuracy disparity [18], demographic parity [16], disparate mistreatment [39] and equalized odds [19]. We use these bias measures because they have been adopted by existing semi-automated bias detection tools (e.g., [8, 32, 37]) for automatically detecting group biases. We developed bias detection prototypes that detect group biases in relation to the four measures.

### 2.2 Semi-Automated Bias Detection Tools

There is a rich ecosystem of semi-automated bias detection tools [3–5, 8, 10, 32, 34, 37]. These tools offer automated support for detecting biases across demographic groups while involving humans in the process of reviewing potential biases. Some of these tools are released as a Python package and do not have a graphical user interface. For example, AI Fairness 360 is an open source Python toolkit that provides a framework and a comprehensive set of bias measures for auditing models and mitigating biases [8]. Tools that offer a graphical user interface often help compute performance measures across demographic groups. Some of them, however, do not identify potential biases and rely on users to identify biases from the computed measures. The What-If Tool, for example, enables users to slice their data into different demographic groups [37]. The algorithm used by the What-If Tool computes performance measures for each group but do not report problematic biases.

We focus on a set of interfaces that automatically report biases based on the computed bias measures (e.g., [5, 32, 34]). After automatically detecting biases, these interfaces often generate a report of detected biases for user review. The automatically-reported biases are commonly presented as a recommendation list. For instance, Themis designs test cases based on user-defined criteria to identify cases of group-based discriminations [5]. These cases of biases are communicated using a recommendation list. Visual cues are also commonly used for reporting automatically-detected biases. Aequitas offers a bias report to show all the computed bias measures and highlight the ones that indicate biases in red color [32]. Our work presents a study to compare these two common styles for reporting automatically-detected biases (i.e. a recommendation list and visual cues). We aim to investigate the behaviors of users as they review bias reports that employ different presentation styles.

## 3 BIAS DETECTION PROTOTYPES[1]

To prepare for the two bias detection prototypes used in the lab study, we first selected a task domain and developed a bias detection algorithm. We then fashioned two prototypes that report the detected biases in two common presentation styles: visual cues and a recommendation list. The prototypes served as research probes [36] for understanding user behaviors and were not intended to suggest new algorithms and interaction techniques. In this section, we describe the task domain, the algorithm, and the prototypes.

### 3.1 Task Domain

Inspired by an increasing use of sophisticated data mining for analyzing employee performance in a corporate setting [13, 30], we designed a fictitious human resource analytics scenario. In the scenario, there were two companies that developed ML models for evaluating employee performance. These models put employees into three classes: `Below` (i.e. below performance requirements), `Meets` (i.e. meets performance requirements), and `Above` (i.e. exceptional performance). Participants in the lab study were asked to imagine themselves to be a consultant tasked with auditing the models using the two bias detection prototypes.

We strove to choose a scenario that might involve fairness concerns and that ML modelers could understand. The employee performance scenario was suited because using historical data to label employees gives rise to potential unfairness. For instance, there may be pre-existing prejudices in past employee reviews. Using past review data to train a model can incorporate human prejudices into the model judgements. Furthermore, we planned to recruit ML modelers from a large technology company. These ML modelers

---

[1]Video demo of the prototypes: https://youtu.be/8ZqCKxsbMHg

had work experience and likely had knowledge about employee evaluation. Being easily understandable, the scenario enabled us to recruit from a larger pool of eligible participants.

## 3.2 Bias Detection Algorithm

We designed a bias detection algorithm based on a typical ML workflow in which a dataset is split into a training set for training the model and a test set for evaluating the model performance. Using the test data, the algorithm computes performance measures for each demographic group and identifies the measures that indicate potential biases. Here, we describe how the two prototypes detect biases from an ML classifier.

**Computing performance measures.** To begin auditing a model, users first select attribute(s) (e.g., `Gender` and `Ethnicity`) to slice the test set into demographic subgroups (e.g., Asian males and Hispanic females). The algorithm then computes performance measures for each group. We consider four common performance measures in bias analysis: classification accuracy, classification rate, true positive rate and false positive rate.

Table 1 shows the statistical definitions of these measures using Asian males as an example. Classification accuracy is computed once for each group while classification rate, true positive rate and false positive rate are class-based measures and are computed for each class. As the employee performance scenario had three classes, the algorithm derived ten measures (classification accuracy + 3 class-based measures × 3 classes) for each demographic group.

**Computing bias measures.** To identify biases in a performance measure, the algorithm computes a bias measure as the difference between the performance measure and a baseline. Following a standard practice [11, 17], the baseline is the same performance measure for a reference group. Based on [11, 17], we use a group's complement as the reference group. For example, the baseline of classification accuracy for Asian males is the classification accuracy for non-Asian males.

A large difference between a performance measure and a baseline may reveal potential group biases. For instance, a significantly lower accuracy on Asian males than on non-Asian males may imply that the model systematically biases against Asian males.

After finding the difference between each performance measure and its baseline (the difference is a bias measure), the algorithm evaluates the p-value for the difference. It then reports the bias measures that have a p-value less than the significance level (set to 0.05 by default). We use statistical significance as a criterion for identifying problematic bias measures as this method has been commonly used in other semi-automated bias detection frameworks (e.g., [34]).

## 3.3 Prototype Using Visual Cues

Based on the employee performance scenario and the bias detection algorithm, we designed two prototypes. The first prototype (Fig. 1) presents all performance measures and highlights the automatically-reported measures using visual cues (Fig. 1c). It supports a suite of interactions to facilitate review of a bias report.

**Selecting groups.** The prototype enables users to select attribute(s) from an attribute list (Fig. 1b) to define demographic groups for further investigation. Upon attribute selection, the main view displays a list of panels, each representing a demographic group. For instance, selecting `Gender` and `Ethnicity` produces panels for Asian males, White females, and so on in the main view.

The prototype color-codes the performance measures in a panel in blue. For each performance measure, it also shows the baseline and the p-value for the difference between the measure and its baseline. For instance, Figure 1c shows that for Hispanic males, the true positive rate for `Above` is 0%. The numbers below the brown dotted line indicate that for non-Hispanic males, the true positive rate for `Above` is 85% and the p-value for the difference is 0.035.

Table 1: The four performance measures supported by the algorithm. Classification, true positive, and false positive rates are class-based measures. We use Asian males and the class `Above` as an example to illustrate their definitions.

| Measure | Definition |
|---|---|
| Classification accuracy | $\dfrac{\text{\# Asian males who are correctly classified}}{\text{\# Asian males}}$ |
| Classification rate | $\dfrac{\text{\# Asian males classified as \texttt{Above}}}{\text{\# Asian males}}$ |
| True positive rate | $\dfrac{\text{\# Asian males correctly classified as \texttt{Above}}}{\text{\# Asian males whose true label is \texttt{Above}}}$ |
| False positive rate | $\dfrac{\text{\# Asian males incorrectly classified as \texttt{Above}}}{\text{\# Asian males whose true label is not \texttt{Above}}}$ |

ML biases often stem from biased training data [9, 18]. For example, a model may have a low classification accuracy on ethnic minorities because ethnic minorities are under-represented in the training data [18]. We conducted informal usability testing and found that users wanted to see statistics about the training data. Based on user feedback, we provide several training data statistics (e.g., sample size of a demographic group in the training data) in the panel and color-code these training data statistics in green to differentiate them from the blue performance measures.

**Highlighting measures.** Performance measures with a statistically significant difference from the baselines are highlighted in pale yellow (Fig. 1c). These highlighted measures are the automatically-reported measures. In Figure 1c, the large deviation between Hispanic males' true positive rate for `Above` (0%) and non-Hispanic males' true positive rate for `Above` (85%) indicates that Hispanic males are much less likely to be correctly classified as `Above`.

**Ranking and filtering groups.** Users can select a performance measure from the `Rank and Filter` by menu (Fig. 1a) to rank and filter the panels. For example, upon selecting classification accuracy, the prototype ranks the panels in the main view based on the difference between classification accuracy and the baseline. It further filters out the demographic groups that do not have a significant difference for classification accuracy.

**Selecting measures.** There is a bookmark button ⚑ next to each performance measure. During the lab study, we asked participants to select the performance measures that warranted further investigation by clicking on the bookmark button.

## 3.4 Prototype Using a Recommendation List

The second prototype (Fig. 2) presents a recommendation list of automatically-reported measures for user review. The reported measures are grouped into four types and are ranked within each group. For a fair comparison with the prototype using visual cues, it enables users to select the name of a demographic group (Fig. 2c) to examine all performance measures (including the ones that are not automatically reported). The tool supports interactions similar to the previous prototype to help review biases.

**Selecting groups.** Users can select attribute(s) from the attribute list to specify demographic groups. The bias detection algorithm computes performance measures for each selected demographic group and identifies the measures that might constitute potential biases. The prototype shows these automatically-reported measures as a list of entries (see Fig. 2 left).

Each entry corresponds to an automatically-reported measure. At the top of each entry is a label that represents the demographic group (Fig. 2c). The leftmost box (with blue text) displays information about the detected bias, which includes the performance measure, its baseline, and the p-value for the difference between the measure and

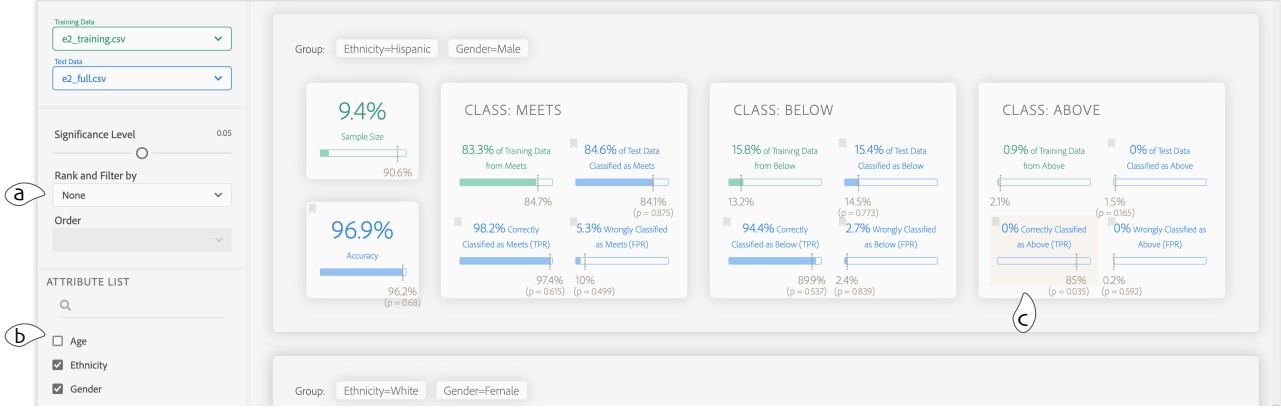

Figure 1: The prototype using visual cues. As the user selects `Ethnicity` and `Gender` (b), it displays panels that correspond to the intersectional ethnic and gender groups in the main view. The top panel represents Hispanic males. The blue performance measure highlighted in pale yellow (c) is an automatically-reported measure. It indicates that for Hispanic males, the true positive rate for `Above` is only 0% while for non-Hispanic males, the true positive rate for `Above` is 85%. The p-value for the difference is 0.035. Users can rank and filter the panels (a).

the baseline. To facilitate interpretation, the top of the box displays a short textual description of the difference.

Similar to the prototype using visual cues, this prototype provides training data statistics for each entry. Based on user feedback from the informal usability testing, the prototype associates each detected bias with relevant statistics about the training data.

**Grouping measures.** The automatically-reported measures are grouped into four types: classification accuracy bias, classification rate bias, true positive rate bias, and false positive rate bias. As an example of how the measures are grouped, the "classification accuracy bias" group contains all accuracy measures that are significantly different from their baselines.

**Ranking and filtering measures.** Within each bias type, the automatically-reported measures are ranked based on the difference between the performance measure and the baseline. When users select a performance measure from the `Rank and Filter by` menu, the prototype only displays the corresponding measures.

**Seeing all measures.** Users can click on a label that shows the demographic group (Fig. 2c) to see the details. A panel similar to that in the first prototype will appear to show all the performance measures (including the unreported measures) for the selected demographic group (Fig. 2d).

**Selecting measures.** There is a bookmark button 🔖 at the top left of a box that shows a performance measure. The panels that appear upon clicking on the demographic group labels also have a bookmark button next to each performance measure.

## 4  LAB STUDY

With the two bias detection prototypes, we conducted a lab study during which participants used both prototypes to review bias reports. This section presents the study details and results.

### 4.1  Participants

We recruited 16 ML modelers (6 female, 10 male, aged 21–37) from a large technology company. We advertised the user study on an intern Slack channel and via internal emails. None of the participants had seen the prototypes prior to the study. Participants had between 1–7 years of experience with ML and all reported experience with training ML models. Some participants (6/16) reported that they had audited their models for biases. While auditing their models, they checked performance measures (e.g., true and false positive rates)

across demographic groups (4/16) and imbalanced data (2/16). We compensated participants with a $25 Amazon gift card.

### 4.2  Procedure

The lab study lasted between 60 and 75 minutes. Participants first watched a tutorial video that introduced the four performance measures supported by the prototypes (i.e. classification accuracy, classification rate, true positive rate, and false positive rate). The tutorial video also explained the corresponding four bias types and their implications. For example, it described classification accuracy bias using the model performance on females as an example: The classification accuracy on females was 81.4% while that on non-females was 92.9%, indicating a bias against females. Participants then watched another tutorial that depicted the interfaces of the prototypes and the interactions supported.

Following the tutorials, they read the employee performance scenario and the task instructions. We asked participants to consider a scenario where a consultant was auditing two ML models. The two models came from two companies that used ML to evaluate employee performance. From each company, the consultant gathered a training set for training the model and a test set with the predictions from the model. To ensure that participants took the tasks seriously, we reminded them of the potential consequences of failing to uncover critical biases. For example, some employees would be unfairly evaluated and the companies' reputation could be harmed.

The task instructions asked participants to audit one model using the first prototype and another model using the second prototype. For each prototype, we instructed participants to review a bias report for intersectional ethnic and gender groups (e.g., Asian males, and Hispanic females). We further asked them to select the performance measures that warranted further investigation by clicking on the bookmark button next to the measures. Our pilot study revealed that some pilot participants took more than two hours to audit a model if we did not put a constrain on the demographic groups for auditing, making the tasks impractical for a short lab study. We decided to focus on intersectional ethnic and gender groups as they are common in evaluating AI systems (e.g., [9]).

After reading the scenario and the task instructions, participants practised using the two prototypes. The bias reports participants reviewed during the practice were different from the ones in the test tasks. During the practice, the experimenter answered participants' questions and helped overcome difficulties.

Following the practice were the two sessions during which partic-

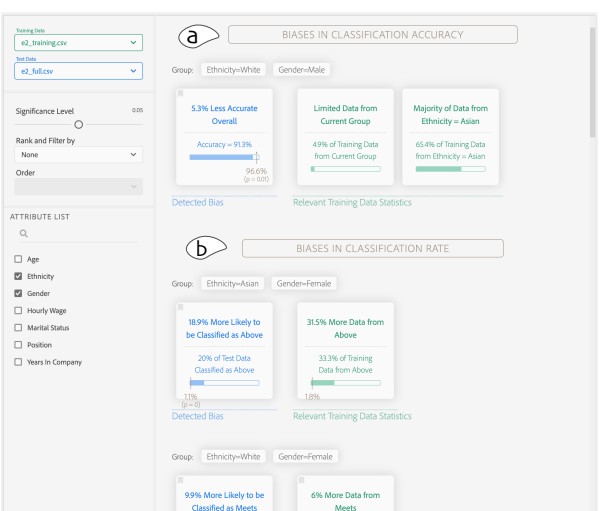
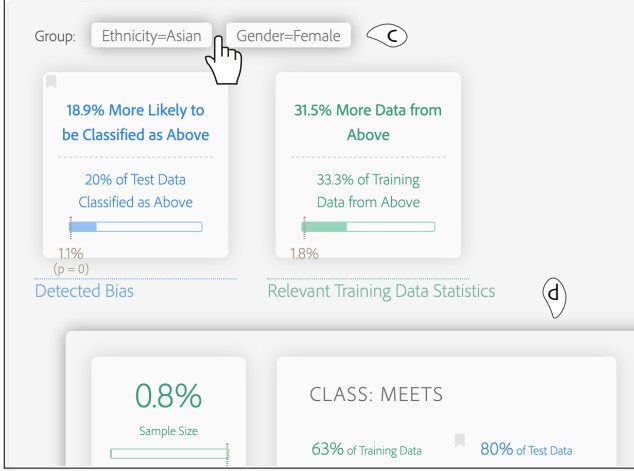

Figure 2: The prototype using a recommendation list (left) and a detected bias (right). As the user selects `Ethnicity` and `Gender`, it shows a list of entries in the main view (left). Each entry corresponds to an automatically-reported measure. The measures are grouped into four types and the headings (a-b) indicate the type of the biases below. As users click on a group label (c), a panel that shows all performance measures for the group (including the unreported measures) appear (d).

ipants used the two prototypes to review biases. At the beginning of each session, they selected `Gender` and `Ethnicity` from the attribute list to focus the bias analysis on the intersectional groups. They started the review task thereafter. We gave participants at most 10 minutes for each prototype. Participants could stop early if they felt that they were done with the task. To mitigate order effects, we counterbalanced the presentation order of the prototypes.

At the end of each session, we asked participants to explain the criteria used for selecting performance measures. To dive into participants' reasoning, we considered asking them to think aloud while reviewing the bias reports. However, thinking aloud while reviewing could interfere with the completion time. After the two sessions, participants rated their preference for the prototypes on a 7-point Likert scale.

### 4.3 Datasets

With each prototype, participants reviewed a bias report for intersectional ethnic and gender groups. The report was generated from a set of training and test data by computing performance measures for all intersectional groups and identifying the measures that indicated statistically significant biases. As each participant reviewed both reports during the study, we needed to ensure that the report contents were different to mitigate learning effects: If the contents of both reports were the same, participants would have applied the knowledge about the performance measures they selected in the first report while reviewing the second report. Hence, while preparing for the two sets of training and test data for report generation, we varied the values of the performance measures across the two reports. To ensure that results from the two sessions were comparable, we further controlled for the number of automatically-reported measures (the algorithm reports measures with a p-value less than 0.05). Here, we briefly describe how we generated the two sets of training and test data for mitigating learning effects across sessions.

We used a modified IBM HR Analytics Employee Attrition and Performance dataset [1] to generate the two sets of training and test data. The modified dataset had 8 features (e.g., `Gender` and `Ethnicity`) and a class label (`Below`, `Meets`, and `Above`). We prepared the two sets of training and test data using a sampling strategy. First, we reshuffled the probability distribution for each feature in the modified dataset. For example, if the probability

distribution of `Ethnicity` is 70% White, 20% African, and 10% Asian, it might become 70% Asian, 20% White, and 10% African after reshuffling. Based on the new probability distributions, we sampled 300 datasets with 5000 records each. To create ground truth labels for the 300 new datasets, we used a random forest model trained on the original modified dataset. We then split each dataset into a training set with 3500 records and a test set with 1500 records. To generate the model predictions, we trained a random forest model on each training set and used the model to label the corresponding test set. Finally, from the 300 sets, we gathered two sets of training and test data. Both sets had 12 performance measures that were automatically reported as having potential biases.

### 4.4 Prototypes

The above procedure ensured that the two bias reports were different (i.e. have performance measures of different values) while showing the same number of automatically-reported measures (12 in this case). Hence, both prototypes provided the same total number of performance measures and the same number of automatically-reported measures. The prototype using visual cues directly displayed all the performance measures while highlighting the ones that were reported by the algorithm. On the other hand, the prototype using a recommendation list showed a list of 12 reported measures and enabled users to see all the performance measures (including the unreported measures) by clicking on a demographic group label to show the detail panel (Fig. 2d). A video demo of the prototypes is available at: https://youtu.be/8ZqCKxsbMHg

### 4.5 Hypotheses

Prior research in recommender systems suggested that users might adhere to the recommended items and consider the alternatives less [21]. The prototype using a recommendation list shows the automatically-reported measures as a list while enabling users to see the unreported measures by clicking on a demographic group label (Fig. 2c). To make sure participants would inspect the hidden measures, we informed participants that clicking on a demographic group label (Fig. 2c) showed all performance measures for the selected group in a panel (Fig. 2d). However, participants might still overlook these measures. We posit that when reviewing a recommendation list, participants will select fewer unreported measures

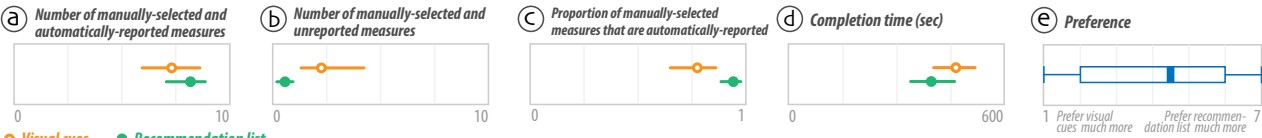

Figure 3: Results for the five quantitative measures. Error bars show 95% bootstrap confidence intervals.

and select mostly automatically-reported measures.

**H1.** Compared with the prototype using visual cues, participants select fewer unreported measures when using the recommendation list prototype to review biases.

**H2.** Compared with the prototype using visual cues, a larger proportion of manually-selected measures (i.e. measures selected by participants) are automatically-reported measures when participants use the recommendation list prototype to review biases.

The prototype using a recommendation list shows less information unless users click on the demographic group labels to see the unreported measures. We hypothesize that participants will complete the task faster given a recommendation list.

**H3.** Compared with the prototype using visual cues, participants complete the task faster when using the recommendation list prototype to review biases.

### 4.6 Measures

We considered the following measures.

**Number of manually-selected and automatically-reported measures.** For both prototypes, we counted the number of measures manually selected by participants that were also automatically-reported. Both prototypes showed 12 automatically-reported measures (as a list in the recommendation list prototype and highlighted in the visual cue prototype).

**Number of manually-selected and unreported measures.** To investigate if participants selected fewer unreported measures when reviewing a recommendation list (**H1**), we counted the number of measures that were manually selected but were not automatically reported. Again, both prototypes showed the same number of unreported measures (shown in a panel upon clicking on a group name in the recommendation list prototype and as measures without highlights in the visual cue prototype).

**Proportion of manually-selected measures that are automatically-reported.** To investigate whether participants adhered to the automatically-reported measures while reviewing a recommendation list (**H2**), we calculated the proportion of automatically-reported measures among the manually-selected measures.

**Completion time.** For **H3**, we measured the completion time for each prototype as the time between participants selected `Gender` and `Ethnicity` from the attribute list and when they told the experimenter that they finished.

**Preference.** In the end-of-study questionnaire, we asked, "Which interface do you prefer for auditing machine learning models for biases?" The scale ranged from 1 (prefer interface A much more) to 7 (prefer interface B much more). Participants further explained their response by comparing the advantages and disadvantages of the prototypes.

**Explanations.** We asked participants to explain their reasons for wanting to investigate the set of selected performance measures. We manually transcribed the verbal data and open-coded the explanations to identify common themes.

### 4.7 Results

For the hypothesis testing, we conducted paired sample t-tests to compare the above measures between the prototypes when the ANOVA assumptions were not violated. In the case of violations, we used the Wilcoxon signed-rank test, which is a non-parametric equivalence of the paired sample t-test.

#### 4.7.1 Number of Manually-Selected and Automatically-Reported Measures

A Levene's test indicated that this measure violated the homoscedasticity assumption. On average, participants selected 7.31 ($SD$=2.89) automatically-reported performance measures using the prototype with visual cues and 8.19 ($SD$=1.87) automatically-reported measures using the prototype with a recommendation list (Fig. 3a). However, a Wilcoxon signed-rank test showed that the difference was not statistically significant ($Z$=1.35, $p$=.195).

#### 4.7.2 Number of Manually-Selected and Unreported Measures

A Levene's test again indicated a violation of the homoscedasticity assumption. Using a Wilcoxon signed-rank test, we found a significant difference in the number of unreported measures selected by participants ($Z$=-2.58, $p$=.014). On average, participants selected more unreported measures using the prototype with visual cues ($M$=2.25, $SD$=2.89) than when reviewing the recommendation list ($M$=0.56, $SD$=0.81), supporting **H1** (Fig. 3b).

#### 4.7.3 Proportion of Manually-Selected Measures That Are Automatically-Reported

The homoscedasticity assumption was violated based on a Levene's test. A Wilcoxon signed-rank test suggested a significant difference in the proportion of measures that were automatically reported among the ones that were manually selected ($Z$=2.93, $p$=.002). On average, 77.5% ($SD$=22.0%) of the manually-selected measures were automatically reported when using the prototype with visual cues, and 94.1% ($SD$=8.84%) manually-selected measures were automatically reported when using the prototype with a recommendation list (Fig. 3c). Participants appeared to adhere to the automatically-reported measures when reviewing a recommendation list. The result supported **H2**.

#### 4.7.4 Completion Time

When using the prototype with visual cues, participants spent 7.73 minutes ($SD$=2.01 minutes) on reviewing the bias report. When using the other prototype, participants spent 6.58 minutes ($SD$=2.23 minutes) (Fig. 3d). To correct for positive skewness, completion times were log-transformed before the hypothesis testing. A paired sample t-test indicated that the difference was not statistically significant ($t$(15)=-1.51, $p$=.152). The result failed to support **H3**.

#### 4.7.5 Preference

On a scale from 1 (i.e. prefer the prototype using visual cues much more) to 7 (i.e. prefer the prototype using a recommendation list much more), the median rating was 4.5 (Fig. 3e). Seven participants

preferred the prototype using visual cues more while eight preferred the other prototype more. One was neutral.

Some participants commented that the prototype using visual cues enabled easier comparison across groups and classes by showing all measures. P8 said, *"You get all the measures combined together and then you can choose which one you want."* However, some felt that showing all the measures made the model auditing overwhelming: *"It [the prototype using visual cues] shows all the information and you don't know which one you should focus on"* (P1).

On the other hand, they commended the prototype using a recommendation list for being easier to interpret by presenting less information: *"In A [the prototype using visual cues], each group you had 14 metrics. Focusing on 14 metrics together require attention"* (P2), and *"I think it [the prototype using a recommendation list] made it easier to focus on biases one by one"* (P12).

### 4.7.6 Explanations

After participants used each prototype, we asked them to explain the criteria used for selecting performance measures. In general, participants cited a large difference from the baselines and a low p-value as reasons for selecting automatically-reported measures. However, participants also selected unreported measures and provided various reasons for doing so. Here, we summarize the considerations for selecting unreported measures:

**Extreme cases.** Some participants were cautious about extreme measures (e.g., false positive rate for `Below` = 100% and classification rate for `Above` = 0%) and would select these measures even though they were not automatically reported.

**Comparison across groups.** Participants often compared the difference between a measure and its baseline across demographic groups to find the largest difference. They often selected such a measure despite a small absolute difference from its baseline.

**Consequences of biases.** P4 commented that wrongly classifying employees as `Below` could be particularly *"harmful."* P4 often selected measures that were not automatically reported but might imply serious consequences.

**Training data.** Aside from the algorithm's suggestions, participants also gathered evidence from the training data to determine whether a bias was systematic. For example, P14 saw a low classification accuracy that was not automatically reported. However, he observed that the demographic group was under-represented in the training data and bookmarked the low classification accuracy because the small sample in the training data might imply a potential bias against the under-represented group.

## 5 DISCUSSION

The impact of interface design on human-in-the-loop bias detection has been an under-examined issue in the realm of algorithmic bias [26]. Our results offer evidence that interface design matters—it might affect what biases users find and how users detect biases. In this section, we summarize the study results and identify their implications for designing semi-automated bias detection tools. Finally, we reflect on the study limitations.

### 5.1 Information Load and Comprehensiveness

During the study, participants often selected unreported performance measures for further investigation. Some explained that they selected such measures because they were wary about the potential consequences of missing the measures. This implies that unreported measures can still be relevant as they may indicate group biases that a bias detection tool fails to detect. Developing bias detection algorithms that automatically report *all* performance measures users care about is challenging because of the inherent difficulty in operationalizing algorithmic biases. For example, some participants selected unreported measures because of the potential consequences these measures hinted at. Designing methods to identify these measures is tricky because modeling the concept of consequences mathematically is difficult. Hence, bias detection algorithms are imperfect and often miss biases users care about. To avoid missing critical biases, users may want to look at a performance measure even if it is not automatically reported.

The study results further revealed that participants selected fewer unreported measures when using the prototype with a recommendation list. A plausible explanation is that participants gave less consideration to such measures with a recommendation list. The prototype with a recommendation list enabled users to click on a demographic group name to see the unreported measures. During the study, we reminded participants that they could click on a group name to examine and select such measures. Nevertheless, participants still selected unreported measures less with a recommendation list. Using a recommendation list, therefore, may lead to a lack of comprehensiveness in bias detection: Although the unreported measures can be relevant, an interface using a recommendation list can cause users to consider these measures less when compared with one that employs visual cues.

Besides comprehensiveness in bias detection, the study provides evidence for information load being a potential trade-off between the two prototypes: The prototype using a recommendation list employs filtering to reduce the number of performance measures displayed while the one using visual cues shows all the performance measures. From participants' feedback, some felt overwhelmed by the large number of performance measures displayed in the visual cue prototype while others commended the recommendation lists for revealing less information at once.

The study results imply that interfaces using a recommendation list might hamper comprehensiveness as users tend to consider unreported measures less whereas interfaces using visual cues might increase information load if they display all performance measures at once. Therefore, information load and comprehensiveness constitute two important considerations in choosing what presentation style to adopt in semi-automated bias detection tools. In the following, we explain how the comprehensiveness and information load axes characterize a bias detection task and can be used to reason about which presentation style to employ when creating bias detection tools.

The comprehensiveness axis depicts the extent to which a bias detection task requires uncovering all potential biases. A comprehensive review of biases is required in some domains because of legal requirements. For example, the Fair Housing Act in the US stipulates that financial institutions cannot approve or reject mortgage loans based on individuals' protected characteristics such as gender and ethnicity [28, 38]. Developers of ML models that scan through loan applications may want to uncover all potential biases to avoid a violation of the law. In low-stake domains, however, missing biases in ML models may not constitute a significant issue. For example, a small school may employ ML models for grading assignments. The developer may put less emphasis on comprehensiveness when detecting biases in such models.

The information load axis, on the other hand, describes the number of performance measures and demographic groups that are reviewed in a bias detection task. ML modelers often consider the intended use of a model and the context of model deployment when deciding what performance measures and demographic groups to analyze during bias detection [27]. For example, detecting biases in a facial recognition model may involve analyzing the model performance across age groups, genders, and skin types [9]. The number of demographic groups to analyze can be huge when a bias detection task involves an intersectional analysis (i.e. analysis of intersectional groups such as Black women). Information load of a bias detection task will determine the amount of information being displayed in a semi-automated bias detection tool.

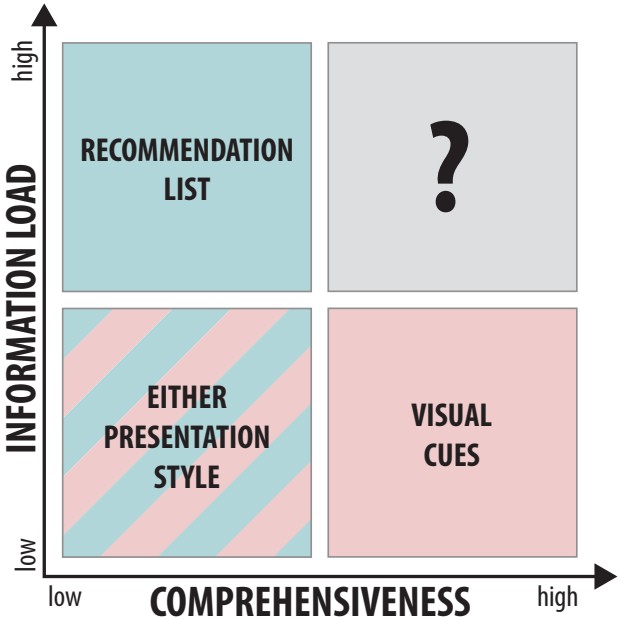

Figure 4: The comprehensiveness and information load axes for characterizing bias detection tasks. Red and blue areas indicate bias detection tasks where respectively using visual cues and a recommendation list may be suitable. The gray area represents an under-examined research area.

Figure 4 shows these two axes and illustrates the situations in which using a recommendation list or visual cues is a suitable choice. The figure does not intend to define precise boundary but aims to provide a guiding understanding of when to adopt which presentation style. The red area corresponds to bias detection tasks that require comprehensiveness but have low information load. Visual cues are suitable for such tasks. Compared with a recommendation list, showing all performance measures while highlighting the automatically-reported ones encourages users to examine all measures including the unreported ones. With the low information load, presenting all performance measures will not overwhelm users. The blue area represents bias detection tasks that have high information load but do not require comprehensiveness. A recommendation list could be applied in these scenarios because they filter out many measures to reduce the information load. Although they may cause users to consider unreported measures less, this may not constitute a problem because comprehensiveness is not a concern. For bias detection tasks that are low in information load and comprehensiveness, either presentation style can be used.

The two axes also reveal areas that are yet to be explored—bias detection tasks with both high information load and a high comprehensiveness requirement (the gray area in Fig. 4). Our results indicate that promoting awareness of unreported measures can be challenging since users gave less consideration to the unreported measures even when the recommendation list prototype provided users the option to look at such biases with a simple click. Future work will investigate techniques that encourage users to examine unreported measures while enabling users to review fewer measures. A promising avenue is perhaps a combination of a recommendation list and visual cues. These tools can utilize algorithms that measure the importance of performance measures. With such algorithms, a bias detection tool can filter out unimportant and unreported measures to reduce information load while showing important and unreported measures to ensure users will look at them more closely. To de-

velop such algorithms, future work will explore what users think are important performance measures in different real-world tasks.

## 5.2 Study Limitations

Our study is limited in that participants only investigated biases against intersectional ethnic and gender groups in relation to a few performance measures. In practice, practitioners may consider a more diverse set of demographic groups and performance measures. While including more demographic groups and measures can yield more realistic results, it is challenging to fit the review tasks within a short lab study—fatigue will be an issue because the tasks will be too long. To investigate user behaviors in a more realistic setting, future work will conduct longitudinal studies during which researchers observe user activities as they adopt a bias detection tool in their typical workflow for weeks or months.

While we observed that participants tended to consider the unreported measures less when reviewing a recommendation list, in the real world, experience may insulate users from the undesirable behavior. For instance, expert auditors may consider the unreported measures more because they are wary of the consequences of missing critical group biases. Nevertheless, while practitioners are experienced in developing ML models, they are often new to the idea of ML fairness [20]. Our results can still have implications for designing tools that target practitioners who have just begun to audit their ML models for biases. However, it is prudent to replicate our study with practitioners who are more experienced in model auditing.

## 6 CONCLUSION

Our paper reported on a small lab study during which 16 participants reviewed bias reports using semi-automated bias detection prototypes that employ different styles for presenting biases. We found that participants often considered further investigating the performance measures that were not automatically reported. However, when using the prototype with a recommendation list, they tended to give less consideration to such measures. Grounded in the findings, we proposed the information load and comprehensiveness axes for characterizing bias detection tasks and discussed the range of bias detection tasks in which a recommendation list or visual cues are a suitable choice of presentation style. The two axes further revealed bias detection tasks with both high information load and a high comprehensiveness requirement as a future research avenue. There has been a lack of research concerning the impact of interface design on human-in-the-loop bias detection. We hope that our study will impel researchers to consider the potential influence of design on how users detect biases and designers to judiciously create interfaces to support users in bias detection.

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
