# OpenReview forum: "The Impact of Presentation Style on Human-In-The-Loop Detection of Algorithmic Bias"
_graphicsinterface.org/Graphics_Interface/2020/Conference — GI 2020_

### Official Review · AnonReviewer2 · 2020-04-20
**Insightful design dimensions for ML bias auditing tools**

**Rating:** 6
**Confidence:** 4

**Review:**

This work investigates interface presentation styles (visual clues and recommendation lists) for auditing group biases in machine learning (ML) models. Through an in-lab within-subject study with 16 ML engineers, the authors evaluate performance measures while using the two types of interfaces. The paper contributes to interface "design dimensions" for bias detection and auditing tasks, i.e., information load, and comprehensiveness.

Creating usable bias detection tools is an important and urgent problem in ML research. I commend the authors for taking on this problem.

Overall this paper is very well written with adequate details about motivation and study design. The related work cites many key papers and does a good job of synthesizing prior literature to situate this work. The findings from this study (design dimensions) offer interesting insights for future research on auditing tools.

However, I do have a few concerns about the design of the interfaces used in the study. I find that overall, the interface design lacks justification about how it affords/supports different types of auditing tasks. For example, I see that end-users need to scroll quite a bit to compare measures across different sub-groups. This may have influenced the number of measures they select. It would also be nice to synthesize insights based on different sub-tasks for bias auditing (even just looking at the "foraging" and "sensemaking" tasks).

Further, the highlighting feature in the visual cues interface is not salient enough (both in the video and the screenshots in the paper are hard to see). And in the recommendation list interface, the "see all" option is not discoverable. As the authors reported, they had to remind participants to click on the group name to see all measures. This confounds the results to some extent.  I would have preferred an *always visible* button instead of clicking on the group header.

The time for each prototype was set at 10 minutes (from a few hours in the pilot). A sentence about this choice might be helpful. Additionally, better measures on cognitive load (e.g., NASA TLX,  CLS questionnaire) could strengthen the study findings.

In summary, while there are some flaws in the study, the results are useful and provide directions for future research. I advise the authors to discuss the above-mentioned limitations of the paper. I recommend accepting this paper for publication.

---

### Official Review · AnonReviewer3 · 2020-04-21
**The paper investigates different visual representations of algorithmic biases and how they affect detection behavior. The study is well motivated and conducted. The paper introduced guidelines for designing bias-detection interfaces and highlights current research gaps in this context.**

**Rating:** 7
**Confidence:** 4

**Review:**

The paper examines different visual representations of algorithmic biases and how they affect the behavior of detection.
Biases of ML algorithms are of increasing interest in the HCI due to a growing number of decision support systems in everyday processes. This problem is well-motivated and sufficiently outlined.

Two prototypes were developed to investigate this bias.
These were well designed and seem sufficient to investigate the problem at hand.
The paper introduces guidelines for designing bias-detection interfaces based on the comprehensiveness and information load necessary. This analysis further reveals current research gaps in the context of bias investigation tools.

Recommendations for minor improvements:
- The introduction as well as the discussion feels in some parts repetitive.
- Figure 1 and 2 are hard to read as a printed version.
- The caption of Figure 1 does not match the figure.
- Typo: 'For example, A model may' (page 4)

Final comments:
The paper is informative, addresses a very timely topic and is well conducted.
The final results open new opportunities for future research.
Hence, I would recommend to accept this paper.

---

### Official Review · AnonReviewer1 · 2020-04-23
**A well-executed study on a timely research topic**

**Rating:** 8
**Confidence:** 4

**Review:**

This paper describes a lab study with 16 participants that investigate the effect of presentation style (recommendation list or visual cues) on user behaviors in reviewing algorithmic bias reports. Through this study, the authors provided guidance in the design of semi-automated bias detection tools.

This paper addresses a timely and important topic. I think the hybrid (qualitative and quantitative) method the authors chose is not the easiest choice but the authors executed it very well. The paper is thoughtfully written and very easy to follow. I also appreciate that the resulting design guidelines can potentially generalize to many critical AI application domains beyond hiring. The outlined design space (Figure 4) could serve as a valuable instrument for designers and researchers working in visualizations and information design for ML outputs.

I do have two critiques, specifically concerning 1) the definition of “algorithmic fairness/bias” and 2) novelty of the findings around information overload/comprehensiveness tradeoff.

I think the paper would benefit from a clearer definition of algorithmic biases. This paper focuses almost exclusively on the *outputs* of algorithmic systems (e.g. accuracy disparity, classification rate, etc.). Algorithmic bias and/or biases in the training data can result in biased system outputs. In this sense, I suspect what the authors meant by algorithmic biases is actually biases in system outputs (including intrinsic biases in training data). Prior work addresses these two kinds of biases quite differently [1].

The other opportunity for improvement is in articulating the novelty of the design guidelines more explicitly. One way to achieve so can be adding a section in the Related Work on related existing data visualization and information design research (e.g. [2] and many more), which could help frame and highlight the novelty of this paper’s findings.

[1] Consider Microsoft’s data card (https://docs.microsoft.com/en-us/powerapps/maker/canvas-apps/working-with-cards) versus modeling card (https://arxiv.org/pdf/1810.03993.pdf)
[2] Designing Theory-Driven User-Centric Explainable AI, CHI’19 https://dl.acm.org/doi/pdf/10.1145/3290605.3300831

---

### Meta-Review · Area_Chair1 · 2020-04-23

**Recommendation:** Accept
**Confidence:** 4

**Metareview:**

All reviewers agree that the paper advanced on a timely topic of algorithmic bias detection. The problem is well-motivated and sufficiently outlined. The paper is very well written with adequate details about motivation and study design.

---

### Decision · Program_Chairs · 2020-04-25

Accept